# The Association between Changes in Built Environment and Changes in Walking among Older Women in Portland, Oregon

**DOI:** 10.3390/ijerph192114168

**Published:** 2022-10-29

**Authors:** Justin Guan, Jana A. Hirsch, Loni Philip Tabb, Teresa A. Hillier, Yvonne L. Michael

**Affiliations:** 1Department of Epidemiology and Biostatistics, Drexel University Dornsife School of Public Health, Philadelphia, PA 19104, USA; 2Urban Health Collaborative, Philadelphia, PA 19104, USA; 3Kaiser Permanente Northwest Center for Health Research, Portland, OR 97227, USA

**Keywords:** older adults, walking, built environment, generalized estimating equations

## Abstract

Some cross-sectional evidence suggests that the objectively measured built environment can encourage walking among older adults. We examined the associations between objectively measured built environment with change in self-reported walking among older women by using data from the Study of Osteoporotic Fractures (SOF). We evaluated the longitudinal associations between built environment characteristics and walking among 1253 older women (median age = 71 years) in Portland, Oregon using generalized estimating equation models. Built environment characteristics included baseline values and longitudinal changes in distance to the closest bus stop, light rail station, commercial area, and park. A difference of 1 km in the baseline distance to the closest bus stop was associated with a 12% decrease in the total number of blocks walked per week during follow-up (e^β^ = 0.88, 95% CI: 0.78, 0.99). Our study provided limited support for an association between neighborhood transportation and changes in walking among older women. Future studies should consider examining both objective measures and perceptions of the built environment.

## 1. Introduction

The built environment can impact one’s health. Various features of the built environment have been shown to improve physical, mental, social health, as well as improve perceived quality of life [1,2]. The built environment can be defined as the environment in which we live and interact in every day [3]. It includes all aspects of the physical environment that are man-made or have been modified by humans, such as access to public transport, commercial areas, and parks, street connectivity, and overall land use mix (the variety of destinations in an area) [4]. Among older adults, one mechanism through which the built environment influences health is by facilitating physical activity. Evidence suggests that subjective or objective measures of the built environment such as walkability, access to recreational facilities, shops/destinations, parks, public transport, intersection density, and street connectivity are positively associated with a variety of measures of walking and overall physical activity among older adults [5,6,7,8,9,10]. Mitra and colleagues found that access to parks and shops are enablers of walking among Canadian older adults [11]. Additionally, Mooney and colleagues found that neighborhood socioeconomic status (SES) was positively associated with walking among older adults, such that as neighborhood SES improves so does walking among older adults [12]. However, most studies examining associations between built environments and walking among older adults have been cross-sectional [5,6,9,13,14]. Thus, evidence for a causal association between the built environment and walking among older adults may be strengthened by additional longitudinal studies. Furthermore, few studies have focused on older women, even though the prevalence of hypertension and obesity is higher among older women compared to older men [15]. Promoting physical activity to combat these conditions among older women is crucial.

To enhance temporal evidence and address gaps in the literature, we aim to examine the associations between baseline values and longitudinal changes in the objectively measured built environment with change in walking among older women by using data from the Study of Osteoporotic Fractures (SOF). Walking is the primary form of physical activity among older adults since it does not require equipment, is low impact, easy to perform, and is free [14]. SOF was a longitudinal study of community-dwelling women recruited at age 65 or older at baseline. We hypothesized that greater distance to the closest bus stop, light rail station, commercial area, and park would be associated with less walking over time.

## 2. Materials and Methods

### 2.1. Analytic Sample

Our analytic sample utilized data from the Study of Osteoporotic Fractures (SOF), a multi-center US cohort study that recruited 9704 community-dwelling (>99% non-Hispanic) white women aged 65 or older between 1986–1988. Women were recruited irrespective of bone mineral density and fracture history; only those unable to walk without assistance or those with bilateral hip replacements were excluded. The study was conducted from 1986 to 2006 and data was collected over nine visits at four sites (Baltimore County, MD, Minneapolis, MN, Mononghela Valley, PA, and Portland, OR) [16]. These years were used because they were the years in which neighborhood information was available. Neighborhood information was not collected after 2006. Participants provided information on demographic characteristics, health history, health behaviors, and psychosocial factors. Measurement and quality control procedures were rigorous (detailed elsewhere) [17] and included a standardized protocol and clinic site training by the SOF Coordinating Center [17]. Details regarding SOF can be found elsewhere [18,19].

These current analyses only included women from the Portland, Oregon site (*n* = 1985). Our analyses focused on the Portland, OR cohort because local planning policies were designed to limit urban sprawl and enhance non-motorized transit in the Portland metropolitan area [20]. The dataset had 20 years and eight visits of follow up available. We excluded individuals with Parkinson’s disease and history of hip fractures because these conditions are highly associated with walking. We also excluded those who lived outside the Portland Metropolitan Area urban growth boundary because quality of built environment data was inconsistent between within and outside the urban growth boundary. Furthermore, we excluded those who were missing data on an indicator of relocating during follow-up because we distinguished changes in the built environment due to relocating to a different neighborhood from changes that occurred within neighborhoods.

Due to of the large size of the cohort, recruitment for each visit was conducted over the span of one or two years. However, not all variables of interest were collected at all visits. To address the unbalanced data and retain as many participants with complete data, we limited analysis to visits 4, 5, and 8, which we used as our baseline or Time_1_, Time_2_, and Time_3_, respectively. Visit 4 (Time_1_ for this analysis) occurred between September 1992–June 1994, Visit 5 (Time_2_) occurred between January 1995–June 1996, and Visit 8 (Time_3_) occurred between January 2002–April 2004. By using these visits, we evaluated 10 years of follow-up and retained potential confounders such as living arrangement, type of home, and self-rated health compared to peers.

### 2.2. Outcome Variable

Total number of blocks walked per a week over time was measured by summing participants’ self-reported number of blocks walked per day for exercise and number of blocks walked per day for their daily routine outside of exercise at every visit. For this analysis, we evaluated the total number of blocks walked per week by multiplying total blocks walked per day by seven. We used total number of blocks walked per week as a measure of total walking, similar to prior related research [21,22].

### 2.3. Exposure Variables

We measured access to neighborhood amenities by computing distances from participants’ residences to the closest bus stop, light rail station, commercial area (such as a shopping center or a mall), and park, based on previous built environment literature [23,24]. We measured the Euclidean distance from participants’ home address (in kilometers), which was geocoded, to the closest bus stop, light rail station, commercial area, and park using administrative data from the Portland Metro’s Regional Land Information System (RLIS) and ArcGIS [25]. Change during follow-up variables were calculated using: $Difference=Time_{i}-Baseline$, where i ranged from 1 to 3 and *Time*_1_ represented baseline. Participants with data at all three time points had three measures of difference, with the difference at baseline having a value of 0 given that baseline values were subtracted from baseline values.

### 2.4. Covariates

Covariates, included self-reported age at baseline (September 1992–June 1994), living arrangement (alone versus with others), type of home (private home/apartment, retirement home, nursing home, personal care home, and other), degree of Activities of Daily Living (ADL) impairment, self-rated health compared to peers (excellent, good, fair, poor, and very poor), smoking status, an indicator of relocating during follow-up, and baseline neighborhood socioeconomic status (SES) [18]. Type of home was categorized as living in one’s own home/apartment and living in other types of homes. ADL impairment variable was calculated from participants’ answers to whether they had any difficulty performing, how much difficulty they had when performing, and whether they receive help from another person when performing the following four activities: walking 2 or 3 blocks outside on level ground, climbing up 10 steps without stopping, walking down 10 steps, and doing heavy housework (like scrubbing floors or washing windows). The total score ranged from 0 to 15, with a higher score indicating more impairment. If participants did not answer a question this variable was not calculated [18]. Self-rated health compared to peers was categorized as good or better and fair or worse. Except age at baseline, all individual-level characteristics changed during follow-up.

Moves/relocations among older adults may indicate a decline in health status given that the overwhelming preference among older adults is to age in place [26]. Thus, changes in neighborhood characteristics due to relocating from one neighborhood to another may affect walking differently than changes in neighborhood characteristics that occurred while participants aged in place. We included an indicator of relocating to a new home during follow-up to adjust for possible differences. We assessed neighborhood socioeconomic status (SES) at baseline based on census tract median household income, median house value, percent earning interest income, percent of residents with a high school diploma, percent of residents with a college degree, and percent of residents with executive management occupations using data from the 1990 Census [27]. Z scores were calculated from these six factors, with a lower total score indicating greater neighborhood deprivation [28]. In our analyses, baseline neighborhood SES scores ranged from −11.02–21.49.

### 2.5. Missing Data

Although the outcome variable (walking) was collected consistently at all SOF visits including baseline, we had missing data for all covariates except baseline age and baseline neighborhood SES; missing was observed for 0.3–33.5% of participants at *Time*_1_, 0.3–36.4% of participants at *Time*_2_, and 0.7–68.3% of participants at *Time*_3_. To address missing data, we used multiple imputation by chained equations using the mice function from the mice package in R [29]. We did not impute missing values due to death or drop out or use these variables in our imputation. To impute variables, all other variables in the dataset were used to predict values using the most appropriate approach. For numeric variables, we used predictive mean matching to impute missing values, which uses observed data to impute missing values. For ordinal variables, we used proportional odds modeling to impute missing values. For binary variables, we used logistic regression to impute missing values. This analysis approach allowed us to retain much of our data because we did not have to drop participants with incomplete data. Instead, we dropped visits for each participant that had incomplete data. This led to participants having a different number of visits available for analysis, with some participants having all three available while others having fewer. Over three-quarters (78.1%) of incomplete cases had missing data due to death and in the original SOF Portland Cohort less than 5% of participants were lost to follow-up. Individuals with incomplete data at all their visits after multiple imputation were also excluded from our analyses.

### 2.6. Statistical Analysis

Baseline characteristics were reported by median number of blocks walked at baseline and for the entire sample. We also compared baseline characteristics for those included in our analyses to those who were excluded. Kruskal–Wallis and Chi-squared tests were used to compare numeric and categorical variables, respectively.

We used generalized estimating equations (GEE), a marginal modeling technique for longitudinal data, to estimate effects of built environment characteristics at baseline and over time on total number of blocks walked per week during follow-up [30]. Since total number of blocks walked per week was a discrete variable, we assumed a Poisson distribution for our models. GEE is used to estimate population average changes in a non-normally distributed outcome in relation to exposures and covariates. GEE is appropriate for unbalanced longitudinal data and provides robust estimates with large samples sizes [30,31]. Several modeling techniques for analysis of longitudinal data were considered. Random effects and generalized linear mixed modeling approaches were not selected because they are most useful to predict subject specific changes or values of an outcome in relation to subject specific values of longitudinal exposures and covariates, for normally and non-normally distributed outcomes, respectively [30]. Additionally, we did not use a fixed effects modeling approach which would have been most appropriate to estimate changes in a normally distributed outcome in relation to longitudinal exposures and covariates because our outcome was a discrete variable [30].

To calculate how walking and each built environment characteristics changed during follow-up we fit the models:$$\log\left(E\left(Walk_{ij}\right)\right)=\beta_{0}+\beta_{1}\left(timerel3_{ij}\right)\;\mathrm{and}\;\;\log\left(E\left(builtenvironmentdifference_{ij}\right)\right)=\beta_{0}+\beta_{1}\left(timerel3_{ij}\right)$$

These models estimate the outcome (walking or a built environment characteristic) using only time as a predictor.

We first fit “univariate models” that had the general form of:$$\log\left(E\left(Walk_{ij}\right)\right)=\beta_{0}+\beta_{1}\left(builtenvironmentdifference_{ij}\right)\;\mathrm{and}\;\;\log\left(E\left(Walk_{ij}\right)\right)=\beta_{0}+\beta_{1}\left(builtenvironmentbaseline_{i}\right)$$

We fit models separately for each built environment distance variable. *Variable_ij_* was the value of *Variable* for the *i*th (*i* = 1,…, *n*) person at visit *j* (*j* = 1,…, *J*) where *J* could be 1, 2, or 3. Time constant variables only had a subscript *i*.

To evaluate potential differences between changes in the built environment due to relocating and changes in the built environment while aging in place, we tested for interactions in the univariate models between the respective built environment variable and an indicator of relocating at some time during follow-up.

To estimate the effects of the built environment characteristics at baseline and over time on total number of blocks walked per week during follow-up we then fit adjusted models using the general form of:$$\begin{array}{cc} \log\left(E\left(Walk_{ij}\right)\right) & =\beta_{0}+\beta_{1}\left(builtenvironmentdifference_{ij}\right)\\ & +\beta_{2}\left(builtenvironmentbaseline_{i}\right)+\beta_{3}\left(timerel3_{ij}\right)\\ & +\beta_{4}\left(timerel3_{ij}*builtenvironmentbaseline_{i}\right)\\ & +Covariate\left(s\right) \end{array}$$

Our data involved a non-continuous function of time, with a gradual change in the trend of the total number of blocks walked per week at three years after baseline (see Figure A1). To address the non-continuous function of time, we created the categorical time *timerel3ij* variable: three years or less after baseline and more than three years after baseline. Additionally, we adjusted for baseline age, living arrangement variables, health related variables, an indicator of relocating during follow-up, and finally baseline neighborhood SES. To evaluate possible over-adjustment, we removed insignificant covariates (α ≤ 0.05) from the model adjusting for all covariates. Results from all of our models estimated using GEE are presented as e^β^ values and their corresponding 95% confidence intervals (95% CIs). The e^β^ value for the *builtenvironmentdifference_ij_* in our adjusted models will allow us to identify the association between change in the built environment characteristics and change in walking. The e^β^ value for the *builtenvironmentbaseline_ij_* in our adjusted models will allow us to identify the association between baseline values of the built environment characteristic and change in walking. e^β^ values are interpreted similarly to odds ratios. For example, suppose e^β^ = 1.2 for baseline distance to the closest bus stop. This would be interpreted as: living 1 km further from the closest bus stop at baseline was associated with a 20% increase in the total number of blocks walked per week during follow-up. We used the geeglm function from the geepack package in R to estimate our models [32]. We did not adjust for multiple comparisons because we tested a priori hypotheses rather than post hoc testing [33]. Results with *p* values less than 0.05 are deemed as significant and will be denoted with asterisks.

### 2.7. Sensitivity Analyses

We evaluated the potential error in the measurement of the total number of blocks walked per week and built environment distance variables. We re-ran final adjusted models with new versions of these variables, eliminating extreme values, which may reflect measurement error. If the value for the total number of blocks walked per week was greater than the 95th percentile of the total number of blocks walked per week at baseline, we set the value for the total number of blocks walked per week to the 95th percentile of the total number of blocks walked per week at baseline. If the value of the built environment distance variable was greater than the 97.5th percentile of that built environment distance variable at baseline, we set the value of the built environment distance variable to the 97.5th percentile of that built environment distance variable at baseline. If the value of the built environment distance variable was less than the 2.5th percentile of that built environment distance variable at baseline, then the value of the built environment distance variable was set to the 2.5th percentile of that built environment distance variable at baseline.

In our second set of sensitivity analyses we evaluated an alternative method for imputing numeric data. Random forest imputation can produce less biased estimates when compared to parametric mice methods, such as linear regression, when data are nonlinear [34]. In a third set of sensitivity analyses we evaluated another alternative method for imputing numeric data that was a similar to random forest imputation, classification and regression trees [35]. All statistical analyses were conducted using R version 3.6.3 [36].

## 3. Results

### 3.1. Baseline Characteristics and Longitudinal Trends

We excluded 732 of the 1985 participants, for a final analytic sample of 1253 (63.12% of the original Portland, OR cohort) (Figure A2). At baseline, the median age in the sample was 71 years, most participants walked for exercise (53.0%), and rated their health compared to their peers as good or better (78.3%). The median number of blocks walked per week at baseline was 56.0 blocks. Comparing those who walked at least 56 blocks per week to those who did not, walkers were slightly younger (median = 70.0 years vs. 72.0 years), had slightly lower BMI (median = 25.6 vs. 26.6), were more likely to rate their health compared to peers as good or better (82.8% vs. 73.2%). Walkers were similar to non-walkers in terms of distance to the closest light rail station (median = 4.3 km vs. 4.5 km), park (median = 0.4 km vs. 0.3 km), bus stop (median = 0.2 km), and commercial area (median = 0.2 km) (see results in Table 1).

Compared to those excluded, participants included in our analyses were more likely to rate their health good or better (78.3% versus 42.1%), and more likely to walk for exercise (53% versus 26.9%) (see results in Table A1). Over 10 years of follow-up, the total number of blocks walked per week on average decreased 37% (e^β^ = 0.63, 95% CI: 0.56, 0.70). Additionally, distance to amenities decreased indicating improvements in the built environment during follow-up. For example, the distance to the closest bus stop and light rail station decreased 5% and 54%, respectively (e^β^ = 0.95, 95% CI: 0.92, 0.97 and e^β^ = 0.46, 95% CI: 0.37, 0.58) (see results in Table 2).

### 3.2. Model Results

In univariate models, change in distance to the closest bus, light rail, and park were positively associated with change in number of blocks walked per week over the follow-up period; only baseline distance to the closest bus stop was significantly associated with change in number of blocks walked per week over the follow-up period (see results in Table 3). Interactions between relocating during follow-up and changes in neighborhood characteristics were nonsignificant, thus we did not stratify multivariable models by relocation status (see results in Table A2).

In adjusted models, associations between baseline and change in neighborhood characteristics with change in the total number of blocks walked per week were generally not statistically significant (see Table 3). While nonsignificant, adjusted estimated indicated that as distances to neighborhood amenities increased over follow-up the number of blocks walked during follow-up decreased. After fully adjusting for all covariates (Model 3), only baseline distance to the closest bus stop was significant. Living 1 km further from the closest bus stop at baseline was associated with a 12% decrease in the total number of blocks walked per week during follow-up (e^β^ = 0.88, 95% CI: 0.78, 0.99). These associations were unchanged after removing insignificant covariates (see Table A3 in Appendix B).

## 4. Discussion

In this study of older white women living in a mid-sized city, we found that as distances to neighborhood amenities increased over follow-up (including distance to public transport, commercial areas, and parks) the number of blocks walked during follow-up decreased, however the associations were generally not statistically significant. Relocating during follow-up was not significantly associated with change in total walking. We will discuss our results in comparison to previous results with regard to: built environment findings, sample differences, and longitudinal design studies.

Few longitudinal studies have evaluated the association between neighborhood characteristics and physical activity. A review by Rosso and colleagues, found that higher street connectivity leading to shorter pedestrian distances, street and traffic conditions such as safety measures, and proximity to destinations such as retail establishments and parks were positively associated with mobility [13]. However, only one of the 37 studies included in the review (2004–2010) was longitudinal. In one longitudinal evaluation, Michael and colleagues found that in a cohort of 422 older men living in Portland, Oregon, GIS measured proximity to parks was associated with greater odds of maintaining or increasing self-reported walking time during follow-up. They adjusted for many covariates, including health behaviors and demographics [37].

Since that time, four additional reviews have examined associations between neighborhood characteristics and physical activity [5,6,13,14]. Only one new longitudinal study was examined by these reviews, and it focused on older adults but did not conduct the analysis separately for women. Gauvin et al. found that closer baseline proximity to 16 GIS measured amenities within walking distance (for example, recreational facilities, grocery chains, shopping centers, and parks) was associated with a greater likelihood of frequent walking (walking 5–7 days a week) in 521 older adults in Canada during 3 years of follow-up. Their models adjusted for perceptions of the built and social environment along with other covariates [38]. Even fewer studies have evaluated change in the built environment in relation to change in walking among older adults. One study found that an increase in the density of recreational facilities within a mile of one’s home was associated with a smaller decline in physical activity. This association was more pronounced among older adults [39]. Similar to our results, another study including middle-age and older adults (men and women, mean age = 62 years) found that greater distance to the bus stop at baseline was associated with decreased walking for transport, while change in the distance to the bus stop during follow-up was not associated with changes in walking for transport. This study also reported an association between greater portions of land zoned for retail at baseline with increased walking for leisure and transport [40].

Several differences may contribute to inconsistences between our findings and previous research. Unlike prior longitudinal research, we found no association between baseline distance to parks or retail amenities and changes in walking over time. Prior longitudinal research was conducted in men or in a mixed cohort, while our analyses were performed only for mostly white women. The influence of the built environment on physical activity may be different for women compared to men, as Gauvin and colleagues found that sex was significant predictor of reporting frequent walking during follow-up [38]. Additionally, our cohort was considerably older, on average, compared to the cohorts included in prior longitudinal research. Evidence suggests that older adults can have different concerns about the built environment when performing physical activity when compared to other age groups. Some concerns include sidewalk quality and perceived safety from traffic [14]. Thus, it is possible that maintaining the same level of walking has different meanings related to the built environment among the oldest old who live into their 80s compared to younger older adults. Then, some existing studies examined total walking along with the component walking outcomes, walking for transport and walking for exercise [40]. This mismatch in outcomes examined may have contributed to differences in our results. Furthermore, our analysis was done in Portland, Oregon, a city known national for its sustainability and walkability [20]. Compared to previous studies that may have been performed in cities less designed for walkability, participants in our study may have walked more or were more likely to walk due to living in Portland.

Our study had several limitations. The modeling approach could have resulted in biased estimates because of unbalanced data and the missingness in the dataset. The alternative imputation methods we evaluated in our sensitivity analyses resulted in minor random changes to the estimates (see results in Table A4, Table A5 and Table A6), providing reassurance that this limitation did not result in substantial bias in our findings. Furthermore, we did not find important differences comparing the baseline characteristics between the imputed and original data (see results in Table A7). Our physical activity outcome, total number of blocks walked per week, has three key limitations: it is self-reported, the length of a block varies by locations, and we assumed that total number of blocks walked per week is seven time the total number of blocks walked per day. Despite these limitations, total blocks walked has been used in prior research as a measure of walking providing some evidence of scientific validity and increasing our ability to compare our findings with prior research [21,22]. Additionally, information bias linked with cognitive impairment is unlikely because cognitively impaired women were excluded from the SOF analytic sample. Finally, our analytic sample was comprised entirely of older white women, and thus our results may not be generalizable to other populations. Despite this limitation, our research was designed to address the lack of literature focusing on how features of the built environment affect walking among older women, even with the greater prevalence of some chronic diseases like hypertension among this population.

Our study also had significant strengths. The community-based SOF cohort was recruited in 1986 at age 65 and older and we believe it is an enormous strength to have two decades of excellent follow-up data collection until the cohort was age 85 and older. While this means that last data collection after 20 years of follow-up was 2006, we believe that the data is still relevant to current policy decisions. Furthermore, SOF was a large, high quality cohort study with well validated measures and many safeguards to reduce bias. We focused on features of the built environment important to promoting total walking among older adults based on prior research [13]. Our analyses were longitudinal over a decade ensuring the appropriate temporality of the association between built environment and walking behavior. Our approach controlled for all time invariant confounders and we were able to adjust for a number of time-varying factors. Additionally, we isolated the effect of baseline values and changes in the built environment on changes in walking from the effect of relocating to a new home during follow-up by adjusting our results by an indicator of relocating during follow-up.

## 5. Conclusions

In our study of older white women living in Portland, OR, distances to amenities decreased during follow-up. Change in access to public transportation, parks, and commercial areas was not associated with changes in walking among older women included in this study sample. However, living further from the closest bus stop at baseline was associated with a decrease in the total number of blocks walked per week during follow-up. Our findings were inconsistent with results from previous studies. Future longitudinal studies are needed to further explore these associations. To inform future guidelines for transportation, land use, and community design planning, we need additional longitudinal evidence evaluating how the built and social environment, including perceptions of changes, influence changes in objectively measured and self-reported walking among older adults [41].

## Figures and Tables

**Table 1 ijerph-19-14168-t001:** Baseline characteristics of the sample stratified by number of blocks walked per week at baseline (median total number of blocks walked per week = 56).

	Walked Less Than 56 Blocks per Week at Baseline (*n* = 586)	Walked at Least 56 Blocks per Week at Baseline (*n* = 667)	Total (*n* = 1253)
Baseline age (years) *			
Median [Min, Max]	72.0 [65.0, 96.0]	70.0 [65.0, 89.0]	71.0 [65.0, 96.0]
Does the participant live alone?			
No	300.0 (51.2%)	351.0 (52.6%)	651.0 (52.0%)
Yes	286.0 (48.8%)	316.0 (47.4%)	502.0 (48.0%)
Type of home *			
Not private home/apartment	79.0 (13.5%)	62.0 (9.3%)	141.0 (11.3%)
Private home/apartment	507.0 (86.5%)	605.0 (90.7%)	1112.0 (88.7%)
Body Mass Index (kg/m^2^) *			
Median [Min, Max]	26.6 [16.7, 47.0]	25.6 [15.3, 46.1]	26.0 [15.3, 47.0]
Degree of ADL impairment *			
Median [Min, Max]	1.0 [0.0, 15.0]	0.0 [0.0, 13.0]	0.0 [0.0, 15.0]
Health compared to people their age *			
Fair or worse	157.0 (26.8%)	115.0 (17.2%)	272.0 (21.7%)
Good or better	429.0 (73.2%)	552.0 (82.8%)	981.0 (78.3%)
Does the participant walk for exercise? *			
No	470.0 (80.2%)	119.0 (17.8%)	589.0 (47.0%)
Yes	116.0 (19.8%)	548.0 (82.2%)	664.0 (53.0%)
Does the participant currently smoke?			
No	547.0 (93.3%)	636.0 (95.4%)	1183.0 (94.4%)
Yes	39.0 (6.7%)	31.0 (4.6%)	70.0 (5.6%)
Did the participant relocate before this visit?			
No	518.0 (88.4%)	593.0 (88.9%)	1111.0 (88.7%)
Yes	68.0 (11.6%)	74.0 (11.1%)	142.0 (11.3%)
Neighborhood SES			
Median [Min, Max]	−0.6 [−11.0, 20.2]	−0.3 [−11.0, 21.5]	−0.4 [−11.0, 21.5]
Distance to the closest bus stop (km)			
Median [Min, Max]	0.2 [ 0.0, 6.8]	0.2 [0.0, 4.2]	0.2 [0.0, 6.8]
Distance to the closest light rail station (km)			
Median [Min, Max]	4.5 [0.1, 22.0]	4.3 [0.1, 21.3]	4.4 [0.1, 22.0]
Distance to the closest commercial area (km)			
Median [Min, Max]	0.2 [0.0, 3.1]	0.2 [0.0, 3.5]	0.2 [0.0, 3.5]
Distance to the closest park (km)			
Median [Min, Max]	0.3 [0.0, 1.6]	0.4 [0.0, 1.8]	0.4 [0.0, 1.8]
Total blocks walked per week *			
Median [Min, Max]	14.0 [0.0, 49.0]	126.0 [56.0, 420.0]	56.0 [0.0, 420.0]

* *p* < 0.05.

**Table 2 ijerph-19-14168-t002:** Change statistics of the total number of blocks walked per week and the built environment during follow-up from the “change” models.

	e^β^	95% Confidence Interval
Total number of blocks walked per week	0.63 *	0.56, 0.70
Distance to the closest bus stop	0.95 *	0.92, 0.97
Distance the closest light rail station	0.46 *	0.37, 0.58
Distance to the closest commercial area	0.90 *	0.89, 0.93
Distance to the closest park	0.93 *	0.92, 0.95

* *p* < 0.05.

**Table 3 ijerph-19-14168-t003:** Results for univariate and adjusted models.

	Overall (*n* = 1253)		
	Model 1	Model 2	Model 3
	e^β^ (95% CI)	e^β^ (95% CI)	e^β^ (95% CI)
Change in distance to the closest bus stop (km)	1.17 * (1.03, 1.34)	1.00 (0.86, 1.15)	0.96 (0.84, 1.12)
Baseline distance to the closest bus stop (km)	0.88 * (0.79, 0.98)	0.90 (0.79, 1.02)	0.88 * (0.78, 0.99)
Baseline distance to the closest bus stop * timerel3	-	0.91 (0.61, 1.37)	0.92 (0.65, 1.30)
Change in distance to the closest light rail station (km)	1.03 * (1.00, 1.05)	1.00 (0.98, 1.03)	0.99 (0.97, 1.02)
Baseline distance to the closest light rail station (km)	1.00 (0.99, 1.01)	1.00 (0.99, 1.01)	1.00 (0.99, 1.01)
Baseline distance to the closest light rail station * timerel3	-	1.02 (0.99, 1.04)	1.01 (0.98, 1.03)
Change in distance to the closest commercial area (km)	1.11 (0.95, 1.28)	0.90 (0.76, 1.07)	0.88 (0.73, 1.05)
Baseline distance to the closest commercial area (km)	1.03 (0.90, 1.17)	1.00 (0.88, 1.14)	0.93 (0.82, 1.07)
Baseline distance to the closest commercial area * timerel3	-	0.97 (0.77, 1.22)	0.93 (0.75, 1.16)
Change in distance to the closest park (km)	1.38 * (1.03, 1.86)	0.95 (0.68, 1.34)	0.84 (0.63, 1.19)
Baseline distance to the closest park (km)	1.01 (0.84, 1.22)	0.99 (0.82, 1.20)	1.08 (0.91, 1.28)
Baseline distance to the closest park * timerel3	-	0.85 (0.53, 1.36)	0.85 (0.55, 1.31)

* *p* < 0.05, some 95% CIs appear insignificant due to rounding; Model 1: “Univariate” model; Model 2: Age adjusted model; Model 3: Fully adjusted model that adjusted for baseline age, living arrangement, type of home, BMI, degree of ADL impairment, self-rated health compared to peers, smoking status, relocation during follow-up indicator, and baseline neighborhood SES.

## Data Availability

Data and R code will be provided to readers upon request.

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
