# Peer review of "The Association between Changes in Built Environment and Changes in Walking among Older Women in Portland, Oregon"

_ijerph, 2022, doi:10.3390/ijerph192114168_

Round 1
Reviewer 1 Report
The article is a report on the research process with a detailed description of the results. The discussion compares the results with other results obtained in similar studies, also justifying the methodological shortcomings of the conducted study. Against this background, the last part of the article, that is, the conclusion, clearly stands out. It is perfunctory, it does not summarize the presented results or conclusions from the discussion. These two elements need to be completed.
Author Response
Please refer to the attached Word document.

Reviewer 2 Report
Please review the following aspects in your manuscript:
(1) The study was conducted from 1986 to 2006 and data. Is there more recent data to validate the results of the study? Any reasons, why did not continue the study ?
(2) Please include full descriptive statistics of the data collected and provide a comparison among the datasets from the nine visits.
(3) Add plots with the data distribution. Why type of distribution fits the data better?
(4) It seems that the models described in the paper are independent from each other. What model relates "age" to "environment" and "builtenvironmentdifference", and "Walk"?
(5) It appears that there were a validation phase of the test results. Please confirm.
(6) Why hypothesis test was not not conducted?
(7) Considering the amount of data and test results, the conclusions need to be expanded, in particular, about regarding a comparison among the variables considered in the study.
Author Response
Please refer to the attached Word Document.

Round 2
Reviewer 2 Report
Please add a specific section explaining the limitations of the study due to available data/modeling/ and research scope, and how they will impact the conclusions.
Author Response
Please refer to the attached Word Doc. Thank you.
